

# Effect of napping on a bean bag chair on sleep stage, muscle activity, and heart rate variability

Masaki Nishida[1,2], Atsushi Ichinose[1,2], Yusuke Murata[2] and Kohei Shioda[2,3]

[1] Faculty of Sport Sciences, Waseda University, Tokorozawa, Saitama, Japan
[2] Sleep Research Institute, Waseda University, Shinjuku, Tokyo, Japan
[3] Faculty of Human Sciences, Kanazawa Seiryo University, Kanazawa, Ishikawa, Japan

## ABSTRACT

**Background:** Although ample evidence has demonstrated that daytime napping is beneficial for health and cognitive performance, bedding for napping has not yet been scientifically investigated.

**Objectives:** To explore the effect of a bean bag chair on daytime napping and physiological parameters related to sleep.

**Methods:** Fourteen healthy participants were enrolled within the context of a randomized, single-blind, crossover study to evaluate the effects of a bean bag chair in comparison with those of a urethane chair manufactured to have a similar shape. Electroencephalogram, electromyogram, and heart rate variability were recorded and compared between wakefulness and napping.

**Results:** Electroencephalogram analyses revealed no significant differences in sleep architecture or frequency components; however, a significant decrease was found in electromyogram recordings in the trapezius muscle, which represents the neck region ($p = 0.019$). Additionally, a significant main effect of bedding in the low-frequency/high-frequency ratio ($F[1,20] = 4.314$, $p = 0.037$) was revealed.

**Conclusions:** These results suggest that napping in a bean bag chair may provide a comfortable napping environment involving muscle relaxation and proper regulation of the autonomic nervous function.

## INTRODUCTION

The importance of napping during the day has been demonstrated in recent years. Previous research demonstrated a positive association between taking a nap of appropriate duration and the prevention of cardiovascular disease (*Hausler et al., 2019*), type-2 diabetes (*Makino et al., 2018*), and global health as reflected in human longevity (*Gu et al., 2010*). Napping also helps mitigate homeostatic sleep drive, consequently improving subsequent extensive performance, such as memory (*Nelson et al., 2021*), emotion (*Lau et al., 2020*), and physical performance (*Hsouna et al., 2020*). For shift workers, scheduling naps is a critical issue from a labor management perspective (*Davy & Gobel, 2018*;

Corresponding author
Masaki Nishida, nishida@waseda.jp

*Okamoto, Mizuno & Okudaira, 1997*). Although long and frequent naps have been shown to increase the incidence of cardiovascular diseases (*Hausler et al., 2019*) and mortality (*Bursztyn et al., 1999*) especially for the elderly, it has been commonly accepted that appropriate daytime napping is beneficial for human health and performance, as well as counteracting excessive daytime sleepiness.

Several studies have demonstrated that the overall beneficial effects of napping can be obtained even after a short nap of under 30 min. Naps exceeding 30 min during the day facilitate deep sleep in the form of slow-wave sleep (*Hayashi, Motoyoshi & Hori, 2005*), thereby prolonging sleep onset latency in subsequent nighttime sleep (*Rosekind et al., 1995*). Moreover, waking up from slow-wave sleep causes sleep inertia, reflecting temporary sleepiness, decreased alertness, and impaired performance upon awakening (*Stampi, 1992*). There is a general agreement that short midday napping offers a variety of benefits, encompassing cognitive and executive performance.

Sleep, including daytime napping, is sensitive to the environment, including the bedding and mattress used. An inappropriate sleep environment results in difficulty falling asleep and affects nap quality and recovery from fatigue, and subsequent sleepiness (*Okamoto, Mizuno & Okudaira, 1997*; *Okamoto-Mizuno et al., 1999*). Several studies have scientifically assessed the effect of bedding on nocturnal sleep and physiological variables during sleep (*Chiba et al., 2018*; *Herberger et al., 2020*; *Yu et al., 2020*); a few studies have also demonstrated the relationship between suitable napping and associated environmental factors. *Zhao et al. (2010)* showed that napping in a chair with the trunk tilted forward 45° and the head resting on a unique pillow was not significantly different from a nap in a bed in terms of objective sleep variables; however, compared with no-napping individuals, subjective sleepiness and fatigue improved in both the chair and bed conditions. *Roach et al. (2018)* demonstrated that the quantity and quality of sleep obtained in a reclined position were similar to those obtained when resting in a flat seat; unlike the upright posture, reclined positions diminish psychological arousal through autonomic nerve regulation. Previous studies on bedding for napping have focused on the chair angle, without sufficient consideration of the bedding material.

A bean bag chair (BC) consists of a polystyrene bead filling inside a bag; it is transformed into a supportive chair thanks to an expert ergonomic design. In light of previous studies that have shown that reclining chairs provide an advisable napping environment (*Roach et al., 2018*; *Zhao et al., 2010*), BCs are likely to provide appropriate napping conditions with a comfortable reclining posture, enabling proper regulation of the sympathetic nervous system to ensure the beneficial effects of napping. Moreover, the unique material and reclined posture may also reduce muscle tension related to sleep, presumably resulting in increased relaxation.

In this study, we assessed the effects of a BC on nap propensity, as reflected by physiological indices, and compared the findings with those for a traditional sofa-type chair manufactured with urethane (UR). Although past research showed that UR mattresses improved nocturnal sleep quality as measured by actigraphy, there was no evidence to show a beneficial effect of UR chair on daytime napping. The experiment was carried out according to a randomized single-blind crossover design. The assessments of

napping on each chair were based on sleep polysomnography (PSG), alongside continuous electroencephalograms (EEGs), electrooculograms (EOGs), and electromyograms (EMGs). Between-napping assessments of subjective sleepiness (Karolinska sleepiness scale (KSS)) and sleep quality (questionnaires and visual analog scale (VAS) for the sleep status) were administered. We evaluated the activities of superficial muscles attached to the cervical and lateral abdominal region during naps. In addition, heart rate variability (HRV) was calculated both before and during napping to evaluate the autonomic nervous function. We hypothesized that (1) a BC subjectively improves the quality of napping, (2) preferentially regulates nap propensies as measured by EEG, (3) relieves the tension level of superficial muscles activity, and (4) helps modulate autonomic nervous function to provide proper cognitive and physical recovery.

# MATERIALS AND METHODS

## Ethics statement

The research protocols were aapproved by the Academic Research Ethical Review Committee of Waseda University (IRB #2019-193) and the work was performed in accordance with the 1964 Declaration of Helsinki. Written informed consent was obtained from all participants before participating in experiments.

## Materials

Figure 1 shows the bedding and lying posture used in this study. For the BC condition, a BC filled with soft micro-beads was used (Yogibo Max®; Webshark Inc., Osaka, Japan). For the UR condition, a UR-based, polyethylene-surfaced cushion chair (LOWYA® Long seat chair; Vega Co., Fukuoka, Japan) was used. The chair sizes were quite similar (BC: 170 cm × 48 cm × 70 cm, UR:172 cm × 56 cm × 75 cm), and the BC was 16% lighter than the UR (BC: 8.0 kg, UR: 11.5 kg). The chairs were installed in a quiet and dark shield room in the sleep laboratory of Waseda University, where the temperature was set at approximately 25.0 °C and the light intensity level was adjusted to approximately 200 lx, as measured with a light meter.

## Participants and setting

Fourteen healthy participants (two females and 12 males, age: 22.5 ± 1.5 years) with normal sleep habits were enrolled. The sample size calculation was determined using G*Power 3.1.9.6 statistical power analysis software (*Faul et al., 2007*). Power analysis indicated that a sample size of 12 per group (number of measurement = 2) were needed for a $\eta_p^2$ (0.25) when $\alpha = 0.05$ for a power of 0.60 with two independent conditions, using repeated measures of variance (ANOVA), within -between interactions. Statistical significance was set at $p < 0.05$ (two-tailed).

Participants were recruited through poster advertisements, social media, and emails to students of Waseda University. Participants were selected based on the absence of any prior diagnosis of sleep disorders and them not having taken any medication and alcohol at the time of the experiments. Participants were asked to maintain regular sleep schedule for 1 week before the laboratory experiment.

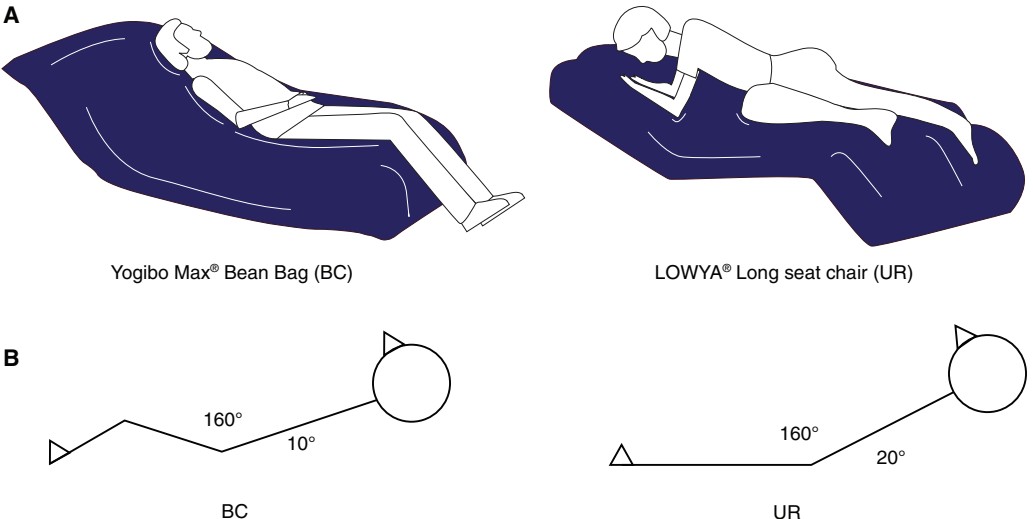

**Figure 1 Illustration of a bean bag chair (A) and schematic view of napping posture from the side in both napping environments (B).** Abbreviations: BC, bean bag chair; UR, urethane chair.

All participants were assessed using the Pittsburgh Sleep Quality Index (PSQI) for habitual sleep quality and Morningness-Eveningness Questionnaire (MEQ) for individual circadian preference. MEQ was a self-assessment questionnaire to determine morningness and eveningness, representing the degree to which respondents are active and alert at certain times of day (*Horne & Ostberg, 1976*). Subjective bedding preferences were also assessed using the visual analogue scale (VAS; higher scores = softer bedding preference). The mean values for the participants' baseline assessment were as follows: body mass index, 22.7 ± 1.8; subjective bedding preferences, 4.9 ± 1.8; PSQI, 5.8 ± 2.8; and MEQ, 43.3 ± 9.6.

## Study design

Participants were enrolled within the context of a randomized, single-blind, crossover study to evaluate the effects of BC and UR at a minimum of 1-week intervals (Fig. 2). Participants entered the laboratory and were asked to rate their level of sleepiness based on the KSS before and after the nap. Participants reported subjective evaluation of sleep in VAS scores after taking a nap.

Participants were not informed about which type of chair they would be napping in. Participants were requested to sit on the experimental chair for napping after the lights were switched off at 13:00. Participants were woken up as lights were turned back on at 14:00 by the examiner.

## Actigraphy recording

To monitor sleep in the night preceding the experiment, participants were requested to wear an MTN-220 (Acos Co., Ltd., Nagano, Japan) on the front side of the trunk by clipping it to their waist belt or the edge of their trousers/pants. This actigraph is a small and light (9 g) coin-shaped device (external dimensions of 27 mm in diameter and 9.8 mm

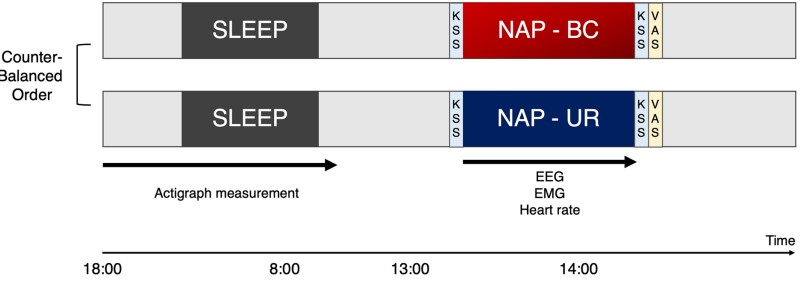

**Figure 2 Experimental design.** The study was conducted in a randomized crossover design, involving a bean bag chair condition (BC) and urethan chair (UR) condition with at least a 1-week interval between the experiments. To monitor sleep in the night preceding the experiment, participants were requested to wear and activate waist actigraphy at 18:00, as well as turn off and remove at 8:00. Participants visited the laboratory at noon, for an explanation of the experiment and preparation of equipment. Participants were requested to sit on the experimental chair for napping after the lights were switched off at 13:00. Participants were woken up as lights were turned back on at 14:00 by the examiner. Participants were asked to rate their level of sleepiness based on the KSS before and after the nap. In addition, participants reported subjective evaluation of sleep in VAS scores after taking a nap. KSS, Karolinska sleepiness scale; VAS, Visual analogue scale; BC, bean bag chair; UR, urethane chair.

in depth, including the clip) that records the amount of physical activity by employing an internal three-axis accelerometer, enabling sleep evaluation at home prior to the experiment. Recording started at 18:00 the preceding night and stopped at 08:00 on the day of the experiment. The actigraph was removed at 8:00 when the recording ended and was not used for the napping experiment. The sensitivity and specificity of MTN-220 were equivalent to those determined for conventional actigraphy (*Nakazaki et al., 2014*) and PSG (*Enomoto et al., 2009*). Nocturnal sleep propensity on the night before the experiment was compared between the two bedding conditions.

## Electroencephalogram and sleep stage scoring

EEG recordings were obtained during the nap using a portable recording system (Brainwave Sensor, ZA-X®; Proassist, Ltd., Osaka, Japan). This system consisted of a wireless transmitter and receiver, using a pair of bipolar EEG and EOG electrode leads connected to the transmitter. The transmitter is 65 mm (W) × 36 mm (L) × 16 mm (H) and weighs 22 g, while the receiver is 135 mm (W) × 76 mm (L) × 27 mm (H) and weighs 135 g. Disposable electrodes (Vitrode F-150M; Nihon Koden Corp., Tokyo, Japan) for channel 1 were placed 1 cm above and lateral to the outer canthus of the left eye and right mastoid. Two other disposable electrodes connected to channel 2 were placed 1 cm below and slightly lateral to the outer canthus of the right eye as well as above the chin.

The signals were recorded at a sampling rate of 128 Hz with filters of 0.5–40 Hz for EEG and 0.5–10 Hz for EOG. Sleep stages were scored every 20 s by a clinical professional technologist without knowledge of the interventions according to the American Academy of Sleep Medicine (AASM) manual scoring rules (*Berry et al., 2020*). Although the 20 s epoch was shorter than that officially defined in AASM scoring rule, applying a shorter epoch provides more accurate validity in determining sleep stages (*Schulz, 2008*). Sleep stages were classified into the four sleep stages: (1) awake, (2) rapid eye movement (REM)

sleep, (3) light non-REM (NREM) sleep (stages N1 and N2), and (4) deep NREM sleep (N3). To validate, simultaneous sleep recordings using the portable two-channel device showed robust agreement in sleep scoring with PSG; for example, kappa values were 0.80 overall and inter-scorer concordance rates were 60.1% in the two-channel and 71.7% in the PSG (*Nonoue et al., 2017*).

EEG recordings during sleep were classified into 20-s intervals and brain waves were categorized as follows: delta (1.0–4.0 Hz), theta (4.0–8.0 Hz), alpha (8.0–12.0 Hz), and beta (16.0–35.0 Hz). Fast Fourier Transform was performed to provide the mean spectral density on extracted intervals to assess the component of each frequency band.

## Electromyography

Continuous EMG recording of body muscles during the nap was carried out, minimizing the number of muscles attached to the channel to avoid disturbing sleep. The muscle activity of the bilateral trapezius muscle and the bilateral abdominal oblique external muscle were measured. Two EMG probes were positioned 2 cm apart on the bilateral external oblique muscle with Ag/AgCl electrodes. Both electrodes were placed around the center of a line drawn between the inferior borders of the 10th rib and pubis and on the muscle belly under the 10th rib. For the trapezius muscle, probes were placed 2 cm apart on the upper fibers of the trapezius, approximately 2 cm to the lateral inferior direction of the vertebra prominens (seventh cervical vertebra, C7). Additionally, referenced probes were also placed on the left shoulder, superiorly.

For the measurement of EMG signals, a multi-telemeter system (Polymate mini A108; Miyuki Giken, Co. Ltd., Tokyo) was employed. Raw EMG signals were sampled at 500 Hz, with a time constant of 0.01 s, and notch-filtered to erase the 50 Hz frequency interference. For the EMG amplification, the high cutoff filter was set to 300 Hz and the low cutoff filter to 30 Hz to exclude frequency relevant to respiratory and any heart- or motion-associated activity (*Abbaspour & Fallah, 2014*; *Chowdhury et al., 2013*).

To assess the activity of superficial muscles in the EMG signal in the time domain, an integrated EMG (iEMG) was calculated. The iEMG is the mathematical integration of a rectified electrical muscle stimulation signal, representing a method for numerical integration and numerical approximation of definite integrals (*Truong Quang Dang et al., 2012*). Specifically, it is the following approximation:

$$Y = \int_{a}^{b} f(x)dx \approx \frac{b-a}{6}\left[f(a) + 4f\left(\frac{a+b}{2}\right) + f(b)\right]$$

In this formula, a and b are the unit spacing. F(a) and f(b) are the amplitudes of the EMG signal at a and b. The iEMG was computed as an approximation of the cumulative integral of the whole EMG signal amplitude during a 20 s period.

## Heart rate variability

Actiheart 5 monitors (Cambridge Neurotechnology, Cambridge, UK) were used to record heart interbeat interval (IBI) data to evaluate HRV. These are flat and lightweight (10.5 g)

wearable devices for which high levels of intra- and inter-instrument reliability, as well as good validity of measures, have been reported (*Brage et al., 2005*).

Participants were requested to attach the Actiheart monitor on their chest with a fastened belt during their nap. A 15-min period of properly collected data during the sleep period was extracted for analysis by visually screening the raw data. More information on how the ECG IBI artifacts were discarded/removed is necessary. Artifacts in IBI data detected visually were removed and the resulting data were concatenated. Expressed IBI data indicated as invalid data according to the software (Actiheart Software ver.5; Cambridge Neurotechnology, Cambridge, UK) were also excluded from the analysis.

In terms of frequency analysis, the integrated spectral power of the periodic components was calculated to quantify the total spectral power (TP; $\leq 0.4$ Hz; ms$^2$) and the power in the low-frequency (LF; 0.04–0.15 Hz; ms$^2$) and high-frequency (HF; 0.15–0.4 Hz; ms$^2$) bands. The HF normalized units (HFnu = HF/(HF + LF) $\times$ 100) and the LF/HF ratio were also computed. Furthermore, for the time domain analysis, the root mean square of successive differences (RMSSD) between adjacent R-R intervals was also calculated. We applied HFnu and RMSSD as indices for parasympathetic tone and LF/HF, which represents the balance between the sympathetic and parasympathetic systems, reflecting sympathetic dominance when it is a higher value (*Djaoui et al., 2017*; *Laborde, Mosley & Thayer, 2017*).

For the analysis of IBI data and frequency-domain HRV measures during different sleep stages, 3-min artifact-free windows were selected during the nap. The IBI data obtained during sleep were extracted based on the sleep stages classified as mentioned above. Sleep stages data to be extracted were N1 and N2 only; N3 stage sleep was excluded because only some participants showed signs of N3 stage sleep. Additionally, a 3-min pre-nap window was analyzed as the wakefulness period. The IBI data obtained during N1 and wake after sleep onset were excluded from the analysis due to difficulty in isolating the recording as a result of instability.

## Fatigue assessment

To assess the effects of napping on recovery from fatigue, a flicker test was conducted after napping. The flicker test was established to measure the flicker perception threshold, which changes as a result of the level of arousal due to fatigue. We evaluated critical flicker–fusion frequency (CFF), reflective of fatigue and mental workload (*Luczak & Sobolewski, 2005*). To avoid sleep inertia, the flicker test was conducted at least 30 min after awakening from the napping experiment.

## Statistical analysis

For demographic variables, we calculated the mean values and standard deviations. The normality of the distributions was confirmed using the Shapiro–Wilk test. Variables from the preceding night and EEG and EMG during napping were normally distributed.

The variables from the night before the experiment and the results of EEG and EMG compared between the two bedding conditions were analyzed with a paired *t*-test. In addition, effect size statistic (d) was analysed to determine the magnitude of the effect

independent of sample size (*Dankel & Loenneke, 2021*). Differences were interpreted using Cohen's (d) guidelines as trivial (<0.2), small (0.2–0.6), moderate (0.6–1.2), large (1.2–2.0), very large (2.0–4.0), and huge (>4.0) (*Hopkins et al., 2009*). To assess HRV between wakefulness and sleep, a two-way ANOVA (Napping condition [BC or UR] × Sleep status [Wakefulness or Sleep]) with repeated measures was performed for each variable. All calculations were performed using SPSS statistics version 28.0 (IBM SPSS Statistics for Windows; IBM Corp., Armonk, NY, USA).

## RESULTS

### Sleep and related parameters: electroencephalogram analysis

Sleep variables from assessments the night prior demonstrated no statistical difference between the two bedding conditions, showing similar results in each condition of the study (Table 1). Actigraphy data of four participants were excluded due to poor recording. The EEG data of one participant were unamenable to analyses because of poor recording as a result of unstable electrode placement. Table 2 presents sleep variables during napping, alongside nap-related parameters. With regard to sleep architecture, no significant differences were found between the two bedding conditions. Seven and six participants demonstrated N3 stage sleep in the BC and UR conditions, respectively. Three participants demonstrated REM sleep in both conditions. In addition, a CFF revealed no significant difference between the two bedding conditions ($t = -0.056$, df = 13, $p = 0.779$, $d = 0.049$). A spectral EEG analysis during each sleep stage showed no significant differences in any frequency band between the two bedding conditions (Table S1).

### Electromyogram analysis during napping

Figure 3 shows the difference in surface EMG activity during sleep in each designated muscle alongside the grand mean of the two muscles. The EMG data of one participant were excluded from the analysis owing to poor EMG recording. The trapezius muscle demonstrated significantly lower EMG activity in the BC condition than in the UR condition ($t = -2.276$, df = 12, $p = 0.019$, $d = 0.764$). The external oblique muscle did not yield a significant difference in EMG activity between the BC and UR conditions ($t = -1.493$, df = 12, $p = 0.093$, $d = 0.464$). The grand mean of EMG activity across participants in the BC condition was significantly lower than that in the UR condition ($t = -2.807$, df = 12, $p = 0.028$, $d = 0.693$).

### Heart rate variability analysis during napping and pre-nap wakefulness

Table 3 shows the HRV variables for each bedding condition between pre-nap wakefulness and NREM sleep during the nap experiment. Similar to the EMG recordings, the HRV data of three participants were excluded from the study owing to poor recording. The panels of Fig. 4 (one panel for each HRV variable) demonstrate the average values for each bedding condition recorded in pre-nap wakefulness and NREM sleep during napping. For all variables, there were no significant interactions between bedding condition and level of awareness. We found a significant main effect of bedding type for LF/HF ($F[1, 20] = 4.314$, $p = 0.037$, $\eta_p^2 = 0.375$), representing a lower LF/HF value in the BC condition than in the

**Table 1 Sleep parameters from the preceding night and subjective scale scores for each bedding condition.**

| Variable measured | BC (n = 14) | UR (n = 14) | p | d |
|---|---|---|---|---|
| Total time in bed (min) | 356 (125) | 324 (113) | 0.152 | 0.287 |
| Total sleep time (min) | 279 (111) | 276 (104) | 0.474 | 0.022 |
| Sleep onset latency (min) | 14.0 (1.0) | 15.0 (1.0) | 0.929 | 0.143 |
| Sleep efficiency (%) | 79.3 (15.0) | 85.0 (11.8) | 0.279 | 0.494 |
| Wake after sleep onset (min) | 50 (52) | 34 (26) | 0.201 | 0.594 |

Note:
BC, bean bag chair; UR, urethane chair; d, effect size. Values are shown as mean (standard deviation) unless specified. No statistical differences were observed in all variables between the two bedding conditions.

**Table 2 Comparisons of nap-related parameters between BC and UR chairs.**

| Parameters | BC (n = 14) | UR (n = 14) | p | d |
|---|---|---|---|---|
| KSS (pre-nap) | 4.2 (1.6) | 4.3 (1.4) | 0.881 | 0.045 |
| KSS (post-nap) | 4.1 (1.4) | 4.5 (2.2) | 0.273 | 0.298 |
| KSS change (%) | 112.7 (70.7) | 122.0 (64.5) | 0.775 | 0.101 |
| VAS | 5.6 (1.3) | 4.4 (2.1) | 0.131 | 0.387 |
| CFF (Hz) | 36.1 (2.4) | 36.0 (1.9) | 0.779 | 0.049 |
| Total time in bed (min) | 71.1 (6.3) | 71.8 (7.4) | 0.701 | 0.099 |
| Sleep period time (min) | 49.6 (7.5) | 50.0 (9.8) | 0.914 | 0.037 |
| Total sleep time (min) | 46.9 (8.3) | 46.7 (10.3) | 0.960 | 0.018 |
| Sleep onset latency (min) | 16.4 (7.4) | 17.0 (6.3) | 0.791 | 0.074 |
| Sleep efficiency (%) | 94.3 (5.7) | 93.7 (8.7) | 0.831 | 0.072 |
| Wake after sleep onset (min) | 2.7 (2.7) | 3.3 (4.5) | 0.708 | 0.122 |
| %N1 | 21.7 (9.9) | 23.5 (15.8) | 0.721 | 0.114 |
| %N2 | 47.7 (19.2) | 44.99 (18.2) | 0.698 | 0.151 |
| %N3 | 17.6 (22.4) | 20.87 (27.6) | 0.660 | 0.119 |

Note:
BC, bean bag chair; UR, urethane chair; KSS, Karolinska sleepiness scale; VAS, visual analog scale (subjective sleep satisfaction); N2, non-REM sleep stage 2; N3, non-REM sleep stage 3; CFF, critical flicker–fusion frequency. Values are shown as mean (standard deviation) unless specified.

UR condition. There were no significant main effects in other HRV-related valuables associated with the level of alertness.

## DISCUSSION

In this study, we evaluated whether a BC would improve physiological indices during napping, relative to those observed during napping in UR bedding. Although sleep architecture and spectral analysis variables showed no significant difference between the two bedding conditions, significantly mitigated muscle activity, especially in the trapezius muscle, was observed in the BC condition relative to that in the UR condition. In addition, the LF/HF ratio during sleep in the BC condition was decreased relative to that during sleep in the UR condition; this suggests that a BC helps modulate balance between the sympathetic and parasympathetic nervous system during napping.

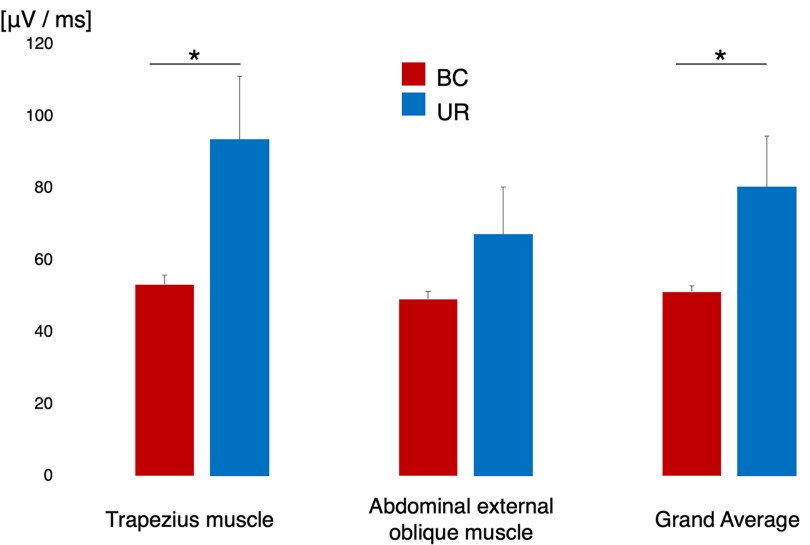

**Figure 3 EMG activity of the trapezius muscle and abdominal oblique muscle, and grand mean of the activity of both muscles across the participants.** The red histogram indicates BC and the blue histogram UR, respectively. An asterisk (*)$p < 0.05$ by paired $t$-test; EMG, electromyogram; BC, bean bag chair; UR, urethane chair.

**Table 3 HRV variables during pre-nap wakefulness and naps in each bedding chair.**

| | BC ($n = 14$) | | UR ($n = 14$) | | Main effect | | | | | | Interaction | | |
| | | | | | Napping condition | | | Alertness | | | Napping condition × Alertness | | |
| | Pre-nap wakefulness | NREM sleep | Pre-nap wakefulness | NREM sleep | $F$(df) | $p$ | $\eta_p^2$ | $F$(df) | $p$ | $\eta_p^2$ | $F$(df) | $p$ | $\eta_p^2$ |
|---|---|---|---|---|---|---|---|---|---|---|---|---|---|
| RMSSD | 90.4 (17.9) | 60.4 (6.4) | 68.6 (10.5) | 76.5 (8.2) | 0.015 (1, 20) | 0.904 | 0.001 | 0.709 (1, 20) | 0.410 | 0.034 | 1.969 (1, 20) | 0.176 | 0.090 |
| 95% CI | [55.2 – 125.6] | [47.7 – 73.0] | [48.1 – 89.2] | [48.1 – 89.2] | | | | | | | | | |
| LF (ms$^2$) | 1,876.4 (548.6) | 1,313.3 (264.3) | 1,594.3 (488.7) | 2,186.5 (475.7) | 1.868 (1, 20) | 0.187 | 0.085 | 0.014 (1, 20) | 0.906 | 0.001 | 1.086 (1, 20) | 0.310 | 0.052 |
| 95% CI | [801.1 – 2,951.7] | [795.3 – 1,831.3] | [636.4 – 2,552.2] | [1,254.1 – 3,188.9] | | | | | | | | | |
| HF (ms$^2$) | 1,790.7 (756.5) | 811.5 (115.1) | 1,212.3 (292.0) | 1,288.8 (200.1) | 0.007 (1, 20) | 0.934 | 0.001 | 0.703 (1, 20) | 0.412 | 0.034 | 1.006 (1, 20) | 0.328 | 0.048 |
| 95% CI | [308.0 – 3,273.4] | [585.9 – 1,037.1] | [640.0 – 1,784.6] | [896.6 – 1,681.0] | | | | | | | | | |
| LF/HF | 1.49 (0.20) | 2.29 (0.43) | 1.75 (0.33) | 2.85 (0.92) | 4.314 (1, 20) | 0.037 | 0.375 | 1.043 (1, 20) | 0.327 | 0.080 | 1.567 (1, 20) | 0.235 | 0.115 |
| 95% CI | [1.10 – 1.88] | [1.45 – 3.13] | [1.10 – 2.40] | [1.05 – 4.65] | | | | | | | | | |
| TP (ms$^2$) | 12,300.2 (4,395.9) | 4,775.5 (845.5) | 5,817.1 (1,225.4) | 7,890.1 (2,210.9) | 0.801 (1, 20) | 0.381 | 0.039 | 2.445 (1, 20) | 0.491 | 0.024 | 0.493 (1, 20) | 0.491 | 0.024 |
| 95% CI | [3,684.2 – 20,916.2] | [3,118.0 – 6,432.4] | [3,415.3 – 8,218.9] | [3,556.7 – 12,223.5] | | | | | | | | | |
| HFnu | 0.44 (0.04) | 0.39 (0.04) | 0.43 (0.03) | 0.40 (0.05) | 0.301 (1, 20) | 0.589 | 0.015 | 3.314 (1, 20) | 0.084 | 0.142 | 0.000 (1, 20) | 0.991 | 0.000 |
| 95% CI | [0.36 – 0.52] | [0.31 – 0.47] | [0.37 – 0.49] | [0.30 – 0.50] | | | | | | | | | |

**Note:**
HRV, heart rate variability; BC, bean bag chair; UR, urethane chair; RMSSD, root mean square of successive differences; LF, low-frequency; HF, high-frequency; TP, total power; HFnu, normalized HF unit; CI, confidence interval. Values are shown as mean (standard error of the mean) unless specified. Statistical significance was evaluated by two-way analysis of variance.

One of the most notable results linked to BC napping was a remarkable decrease in the activity of the trapezius muscle, which is a large triangular muscle extending from the cervical region to the width of the shoulders. The trapezius stabilizes the shoulder blades

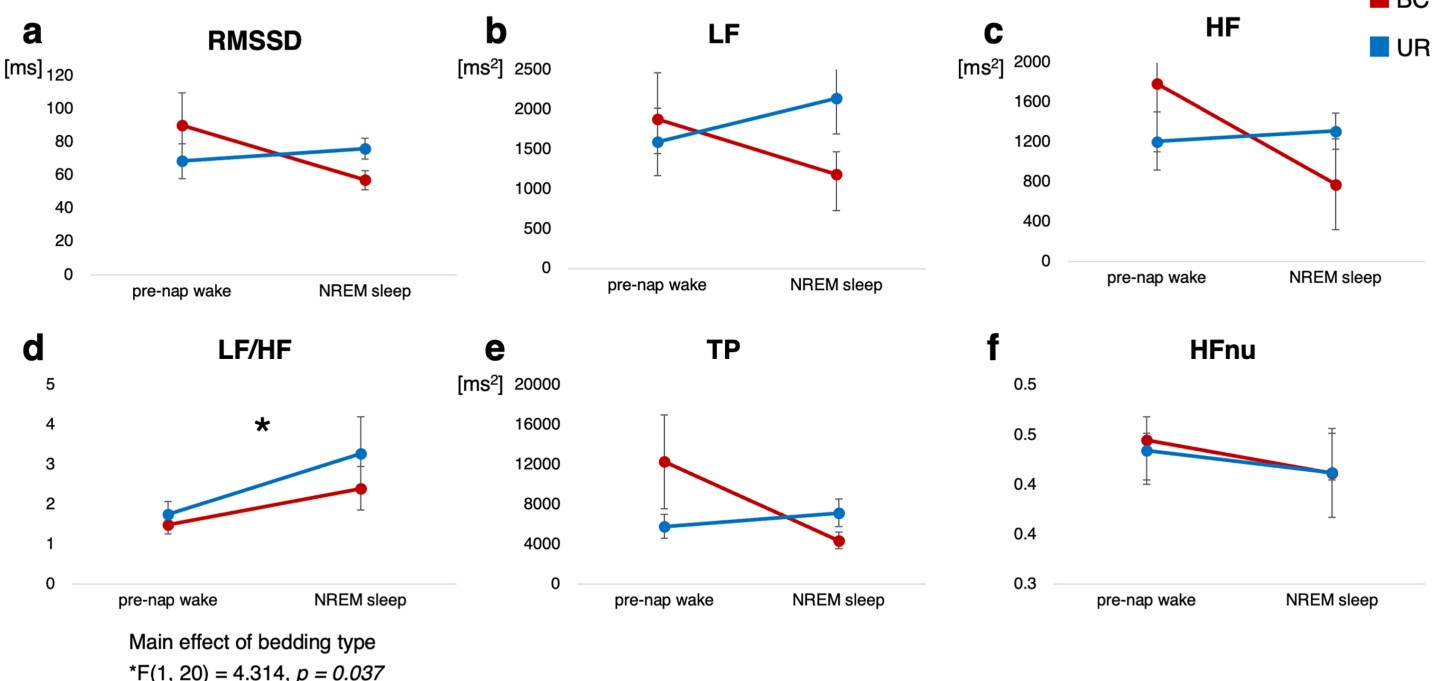

**Figure 4 Change in heart rate variability variables across alertness for each bedding environment.** Red histogram indicates BC and blue histogram UR, respectively. Values in the panel indicate main effect of bedding. RMSSD, root mean square of successive differences; NREM, non-rapid eye movement; LF, low-frequency; HF, high-frequency; TP, total power; HFnu, normalized HF unit.

and facilitates shoulder and neck movements (*Bakkum & Cramer, 2014*). Previous reports have demonstrated that the middle trapezius muscle demonstrates the greatest activation when an individual is laying on their side with a pillow of the least comfortable height (*Sacco et al., 2015*). Moreover, the use of uncomfortable pillows with increased neck tension results in regular waking and poor sleep quality (*Gordon & Grimmer-Somers, 2011*). In contrast, a less significant decrease in muscle activity in the abdominal oblique muscle, which governs ipsilateral side-bending and contralateral rotation of the lower trunk, was seen. Although the UR material itself has possibility to achieve lower distribution of pressure (*Low et al., 2017*), a BC has the potential to provide a napping posture that reduces the muscular load on the neck, possibly preventing neck pain, poor neck mobility, and stiff shoulder.

Among HRV variables, a significant main effect of bedding was observed regarding LF/HF ratio in the BC compared with that in UR bedding conditions, thereby indicating that napping in a BC is associated with a decreased LF/HF ratio. Previous studies have demonstrated that sympathetic activity decreases during NREM sleep, especially deep sleep classified as N3 stage (*Cellini et al., 2016*; *Chen et al., 2020*; *de Zambotti et al., 2014*). In contrast, we detected no significant main effect of alertness; however, our results elucidated that the BC condition has the potential to provide suitable napping conditions for recovery, modulating the LF/HF ratio, which is thought to reflect the magnitude of psychosomatic stress (*Kim et al., 2018*; *Sloan et al., 1994*). It remains controversial but is commonly known that the LF/HF ratio reflects balance between the sympathetic and

parasympathetic nervous system, whereby increases in LF/HF are assumed to reflect a shift in "sympathetic dominance," while decreases correspond to a "parasympathetic dominance" (*Djaoui et al., 2017*; *Laborde, Mosley & Thayer, 2017*). A significant decrease in this autonomic nervous balance with BC use may be attributed to a reduction in the cervical muscle load, adjusting an automatically suitable posture during napping by means of elasticity. Recent imaging study investigating effects of hardness and shape of UR foam mattresses demonstrated that the skin and soft tissue are likely to be affected by the shape of the UR mattress, not by its hardness (*Kumagai et al., 2019*). Inappropriate cervical tension appears to lead to increased autonomic nervous activity as reflected in the LF/HF ratio (*Santos-de-Araujo et al., 2019*; *Shafiq, McGregor & Murphy, 2014*), suggesting that the diminished muscle tone, especially in the neck region, owing to BC bedding is beneficial for proper regulation of autonomic function for napping.

In contrast to the decrease in EMG activity and HRV changes, effects of a BC on sleep stages were not detected. As this study was limited to a short nap experiment not encompassing a whole night's sleep, significant differences in sleep architecture were difficult to obtain for each bedding condition. Even previous studies exploring the effect of high resistance bedding on all-night sleep revealed no statistical differences in sleep architecture, only an increase in delta power in the first cycle of NREM sleep (*Chiba et al., 2018*). Moreover, admitting that napping is a mode of incomplete sleep, represented by the fact that most participants showed no signs of slow-wave sleep or REM sleep, the lack of significant differences in the quantitative frequency analysis is quite consistent with sleep propensity.

However, this study has several limitations. First, the number of enrolled participants was relatively small; larger numbers are required to detect more significant sleep effects. Additionally, as sex differences are not evenly distributed, alterations in sympathovagal activity due to hormonal imbalances associated with the menstrual cycle need to be taken into consideration (*Schmalenberger et al., 2019*). Second, sleep scoring was performed based on a single EEG derivation instead of conventional PSG. Nevertheless, the same EEG expert completed the final confirmation visually, ensuring the validity of the sleep structure and quantitative EEG analysis. Third, the study was designed as single-blind; even if experimental participants did not closely observe the bedding chair they were using, they may have been able to identify whether the chair was a BC or UR. Furthermore, the study investigated the acute effect of the chair on napping opportunity. Studies are needed to evaluate the changes in physiological indices with the continuous use of BCs. Regardless of these limitations, our results point to the possibility that different types of bedding have a considerable effect on napping and its associated physiology. Further research is warranted considering the physiological and ergonomic aspects of a comfortable nap.

## CONCLUSIONS

In conclusion, although the BC did not alter sleep propensity or frequency components, it may reduce muscle activity in the cervical region in association with the LF/HF ratio, reflective of sympathovagal activity in healthy adults. These alterations were reported to be effective in providing a comfortable napping situation for nappers in terms of

psychosomatic aspects. Therefore, napping in a BC, which appears to improve autonomic parameters, is beneficial for a wide range of applications and strategies, encompassing psychosomatic relaxation, improving afternoon performance, and leading to effective recovery for athletes.

## ACKNOWLEDGEMENTS
The authors are grateful to Shutaro Suyama and Sumi Youn for their persistent help.

### Funding
The research was funded by Webshark Inc., as a sponsored research contract (No. B2R500603701). The funders had no role in study design, data collection and analysis, decision to publish, or preparation of the manuscript.

### Grant Disclosures
The following grant information was disclosed by the authors:
Webshark Inc: B2R500603701.

### Competing Interests
The authors declare that they have no competing interests.

### Author Contributions
- Masaki Nishida conceived and designed the experiments, analyzed the data, prepared figures and/or tables, authored or reviewed drafts of the paper, and approved the final draft.
- Atsushi Ichinose performed the experiments, prepared figures and/or tables, authored or reviewed drafts of the paper, and approved the final draft.
- Yusuke Murata performed the experiments, analyzed the data, prepared figures and/or tables, and approved the final draft.
- Kohei Shioda conceived and designed the experiments, analyzed the data, authored or reviewed drafts of the paper, and approved the final draft.

### Human Ethics
The following information was supplied relating to ethical approvals (*i.e.*, approving body and any reference numbers):
Waseda University of granted Ethical approval to carry out the study within its facilities (Ethical Application Ref: 2019-106).

### Data Availability
The data is available at GitHub: https://github.com/masakinishida/researchdata.git.
## Supplemental Information

Supplemental information for this article can be found online at http://dx.doi.org/10.7717/peerj.13284#supplemental-information.

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
