# Peer review of "Effect of napping on a bean bag chair on sleep stage, muscle activity, and heart rate variability"

_PeerJ, doi:10.7717/peerj.13284_

## Round 0.1 · original submission · Minor Revisions

Dear Authors, three experts in the field revised your manuscript and reported some points that should be considered/addressed while revising your manuscript.

Reviewer 1 ·

Basic reporting

This study evaluated effects of daytime napping comparing bean bag chair and urethan bed with multiple objective testing methods. The result of muscle relaxation and parasympathetic activation of bean bag napping was interesting.
However, the authors should address issues below.

1. In 'Participants and setting', authors employed power analysis. Although I understood that power analysis was employed to calculate sample size for this study, the expression is seemed to be unclear for the readers. Please make it clearer.

2. In Study design, authors should make a figure to explain how they employed this study. They employed multiple testing before and after napping, had testing during the napping and napping was consisted by 2 different types of bedding. This study is very complex. A figure explaining the study design will be very helpful for readers.

3. In Electroencephalogram and sleep stage scoring, authors noted that they did sleep stage scoring according to AASM rules. However, I believe that AASM manual recommends 6 EEG plus EOG and EMG with Mastoid reference, sleep stage 1,2,3,REM detection, 30s epoching. Therefore the method of scoring should be different from AASM's. The authors should clarify how they scored. And they noted that their method is validated as Nonoue et al., 2017. If so, they should note that their method is same as Nonoue's.

4. The authors noted that they did not distinguish N1 and N2. However, in table 2, there are N1 and N2 both. Please be consistent.

Experimental design

Experimental design is excellent. Nothing to comment.

Validity of the findings

no comment

·

Basic reporting

The authors use clear, unambiguous and professional English throughout their article. They provide references and sufficient background information. The article is well structured. The layout and content of figures and tables is correct. The necessary research data is supplied. The article is self-contained with relevant results to the research question.

Experimental design

The topic of the article is within the scope of the journal. The research question is well defined, relevant and meaningful. The authors contribute to fill the identified knowledge gap and follow ethical and technical standards. Using the described methods the results can be replicated.

Validity of the findings

The underlying data is shared online and open to the public. The conclusions are well stated and connected to the research question.

Additional comments

The authors provide an excellent scientific article.

·

Basic reporting

General comments:
I reviewed the article “Effect of napping on a bean bag chair on sleep stage, muscle activity, and heart rate variability”. The article aims to evaluate the effect of an bean bag chair compared with a traditional sofa-type chair manufactured with urethane on daytime napping and physiological parameters related to sleep.
Despite the present study indicated that the sleep architecture showed no significant difference between the two bedding conditions, the muscle activity of trapezium decreased and the LF/HF ratio during sleep decreased used an bean bag chair.
The manuscript is interesting and the topic has the potential to be interesting and useful from a scientific point of view. However, additional clarifications about the study design should be added. The main problem is that, in my opinion, is not clear when the authors made the actigraphic monitoring. In some parts of the article seems that is made at home the night preceding the experiment while in other parts seems that is made also during the nap in the laboratory. Clarify this aspect is extremely important to understand the results even if there are no differences for the sleep architecture.
Here below some specific comments.

Experimental design

MATERIALS AND METHODS
Participants and setting
How many participants for each group (BC vs UR)?
Line 124: please report Morningness-Eveningness Questionnaire instead of Morning-Eveningness Questionnaire.
Lines 126-128: specify the units of measure for all the variables.

Actigraphy recording
I don’t understand when the authors made the actigraphic monitoring. Reading the material and methods, I understood that it was made at home the night preceding the experiment while reading the results seems that was made also during the nap in the laboratory.

Validity of the findings

RESULTS
As I said previously, I don’t understand properly the results about sleep and related parameters. In fact in table 1 the authors said to report the comparison between sleep parameters from the preceding night and subjective scale scores for each bedding condition but I saw only the comparison between actigraphic sleep parameters in the two bedding conditions. Where were the sleep parameters of the preceding night? Where were the subjective scale scores? To obtain the actigraphic sleep parameters for the two bedding condition the authors used the actigraphy also during the nap in laboratory? If yes, they have to declare it in the study design. In addition, in the caption as well as in the description of the table 1 remove “No statistical differences…..” because was a repetition of the results and add the numerosity of the participants for each group (BC and UR).
Table 2 the same problem because there were the actigraphic sleep parameters for the two bedding conditions and the authors didn’t report that the actigraphic monitoring was made also during the nap in the laboratory. In case, add it. In addition, in the caption as well as in the description of the table 2 remove “Sleep efficiency-TST/SPT”, add the numerosity of the participants for each group (BC and UR), add units of measure when missing. Add in the description of the table 2 N1 and change in table 2 Flicker into CFF.
Table 3 add units of measure when missing and the numerosity of the participants for each group (BC and UR). Change type into napping condition.
Figure 2 specify that the red histogram was for BC group instead the blue histogram was for UR group and delete the period “The trapezius muscle demonstrates……” because was a repetition of the results.
Figure 3 specify that the red line was for BC group instead the blue line was for UR group and delete the period “Values in the panel……” because was a repetition of the results.

Additional comments

No comments

---

## Round 0.2 · accepted · Accept

Dear Authors,
two experts in the field revised your manuscript and both agreed that it could be published in the present version.

Best regards.
Emilaino Cè

Stefano noted some items that should be addressed at the proofing stage. Please ensure the name of the Institutional Review Board which approved the research, and the IRB approval number, are indicated in the manuscript. Also, the list of abbreviations in the tables should also include the meaning of less common statistical symbols

Reviewer 1 ·

Basic reporting

The manuscript was properly revised.

Experimental design

nothing to point out

Validity of the findings

nothing to correct

·

Basic reporting

No comments to do

Experimental design

No comments to do

Validity of the findings

No comments to do

Additional comments

No comments to do